# A light-gated transcriptional recorder for detecting cell-cell contacts

Kelvin F Cho[1,2], Shawn M Gillespie[1,3], Nicholas A Kalogriopoulos[2], Michael A Quezada[3], Martin Jacko[4], Michelle Monje[3,5,6,7,8], Alice Y Ting[2,9,10,11]*

[1]Cancer Biology Program, Stanford University, Stanford, United States; [2]Department of Genetics, Stanford University, Stanford, United States; [3]Department of Neurology and Neurological Sciences, Stanford University, Stanford, United States; [4]BridgeBio, Palo Alto, United States; [5]Department of Pathology, Stanford University, Stanford, United States; [6]Department of Pediatrics, Stanford University, Stanford, United States; [7]Department of Neurosurgery, Stanford University, Stanford, United States; [8]Howard Hughes Medical Institute, Stanford University, Stanford, United States; [9]Department of Biology, Stanford University, Stanford, United States; [10]Department of Chemistry, Stanford University, Stanford, United States; [11]Chan Zuckerberg Biohub, San Francisco, United States

**Abstract** Technologies for detecting cell-cell contacts are powerful tools for studying a wide range of biological processes, from neuronal signaling to cancer-immune interactions within the tumor microenvironment. Here, we report TRACC (Transcriptional Readout Activated by Cell-cell Contacts), a GPCR-based transcriptional recorder of cellular contacts, which converts contact events into stable transgene expression. TRACC is derived from our previous protein-protein interaction recorders, SPARK (Kim et al., 2017) and SPARK2 (Kim et al., 2019), reported in this journal. TRACC incorporates light gating via the light-oxygen-voltage-sensing (LOV) domain, which provides user-defined temporal control of tool activation and reduces background. We show that TRACC detects cell-cell contacts with high specificity and sensitivity in mammalian cell culture and that it can be used to interrogate interactions between neurons and glioma, a form of brain cancer.

*For correspondence: ayting@stanford.edu

## Editor's evaluation

This manuscript describes engineering of a new system (TRACC) for marking cells that have come into contact with another population of cells. In contrast with previous systems, TRACC is gated temporally and spatially by blue light application. The system comprises a GPCR and ligand that interact at the surface of the two cells, as well as a TEV protease-arrestin fusion that gets recruited following the interaction. The GCPR is fused to a LOV light sensitive domain, a LOV-masked TEV cleavage site and transcriptional activator. TEV cleavage, in the presence of a sender cell and light, releases a transcriptional activator to drive expression of a reporter transgene in the receiver cell. This system provides a new tool for studying cell-cell contacts.

## Introduction

Cell-cell interactions are integral to maintaining cellular and organismal homeostasis; signaling that occurs from direct physical cellular contacts mediates a diverse range of biological processes, including embryonic development, neuronal signaling, and immune-cancer interactions (*Armingol et al., 2021*; *Dustin, 2014*; *Zhang and Liu, 2019*). Consequently, several molecular tools have been developed to visualize and detect cell-cell interactions between different cell populations. Technologies based on

enzymatic labeling strategies result in chemical labeling at contact sites, which allows for direct visualization. For example, LIPSTIC uses sortase A to catalyze labeling on the cell surface of interacting cells and has been applied to study T cell interactions (*Pasqual et al., 2018*). Similarly, the biotin ligase BirA and lipoic acid ligase LplA have been engineered for labeling across synaptic contacts to visualize neuronal synapses (*Liu et al., 2013*). Split forms of horseradish peroxidase (HRP) (*Martell et al., 2016*) and the biotin ligase TurboID (*Cho et al., 2020*; *Takano et al., 2020*) have also been engineered to perform both extracellular and intracellular labeling at cell-cell contact sites; mGRASP (*Kim et al., 2011*) and SynView (*Tsetsenis et al., 2014*) reconstitute GFP across neuronal synapses.

While direct visualization of cellular contacts is useful, it is often desirable to also highlight the *entire* contacting cell, so that cell anatomy, transcriptomic signature, and functional properties can be further characterized. Various tools have been engineered that result in the release of an orthogonal transcription factor (TF) in the receiver cell after contact with a sender cell, allowing for a range of user-desired outputs. These include the Notch-based systems synNotch (*Morsut et al., 2016*) and TRACT (*Huang et al., 2017*), and the GPCR-based system trans-Tango (*Talay et al., 2017*). Other approaches involving trans-cellular uptake of protein cargo have also been developed; in BAcTrace, the botulinum neurotoxin is transferred to the receiver cell, which also results in proteolytic release of a TF (*Cachero et al., 2020*), while in G-baToN, a fluorescent protein is transferred, which labels the receiver cell (*Tang et al., 2020*). These aforementioned tools (TRACT, trans-Tango, and BAcTrace) have not yet been tested in mammalian systems and lack temporal gating, which can provide temporal specificity and reduce background signal (*Kim et al., 2017*).

Here, we describe a different and complementary technology for transcriptional recording of cell-cell contacts. In TRACC (Transcriptional Readout Activated by Cell-cell Contacts), a GPCR in the receiver cell is activated upon interaction with a ligand expressed on sender cells, resulting in the release of a TF, which allows for versatile outputs. By incorporating an engineered light-oxygen-voltage-sensing (LOV) domain, the tool becomes light-gated, and tool activation requires both cell contact and exogenous blue light, restricting activation only to user-defined time windows. We show that TRACC can detect cellular contacts with high specificity and sensitivity in HEK293T culture. We further demonstrate its utility by extending to neuronal cultures and assaying interactions in co-culture between glioma cells and neurons.

## Results
### Design and development of TRACC

To design TRACC, we built upon our previously published tool SPARK, which detects protein-protein interactions with transcriptional readout (*Kim et al., 2019*; *Kim et al., 2017*). TRACC is comprised of four components, as shown in *Figure 1A–B*. On the sender cell, a ligand is presented on the cell surface by fusion to pre-mGRASP, a construct that contains the transmembrane domain of CD4 and the intracellular domain of the pre-synaptic protein neurexin (*Kim et al., 2011*). On the receiver cell, a corresponding GPCR is expressed and fused to an LOV domain, a TEV cleavage site (TEVcs; ENLYFQM), and a TF. Additionally, the receiver cell expresses arrestin fused to the TEV protease (TEVp) and a reporter construct of interest. Upon cell-cell contact, the ligand on the sender cell activates the GPCR on the receiver cell, resulting in recruitment of arrestin-TEVp. However, in the absence of light, the LOV domain cages the TEVcs, rendering it inaccessible to the TEVp. With the addition of exogenous blue light, the LOV domain uncages, resulting in subsequent cleavage and release of a TF and reporter activation. Thus, TRACC is designed as an 'AND' logic gate, simultaneously requiring contact with a sender cell and exogenous blue light.

To first test our design in HEK293T cells, we utilized eLOV, a previously engineered LOV domain that has improved light caging (*Wang et al., 2017*), and the orthogonal Gal4-UAS TF system. To ensure that TRACC would be orthogonal for eventual applications in neuroscience, we selected six GPCR-ligand pairs that are not expressed or lowly expressed in the brain according to the GTEx database (*Lonsdale et al., 2013*). These are CCR3-CCL13, CCR6-CCL20, CCR7-CCL19, GHRHR-GHRH, GNRHR-GnRH, and GCGR-GCG, the last of which was utilized in trans-Tango (*Talay et al., 2017*). GPCR-ligand pairs were cloned into TRACC constructs and co-expressed in HEK293T cells in cis along with a UAS-luciferase reporter. We adopted previously optimized experimental parameters from SPARK (*Kim et al., 2017*) and SPARK2 (*Kim et al., 2019*), including DNA transfection

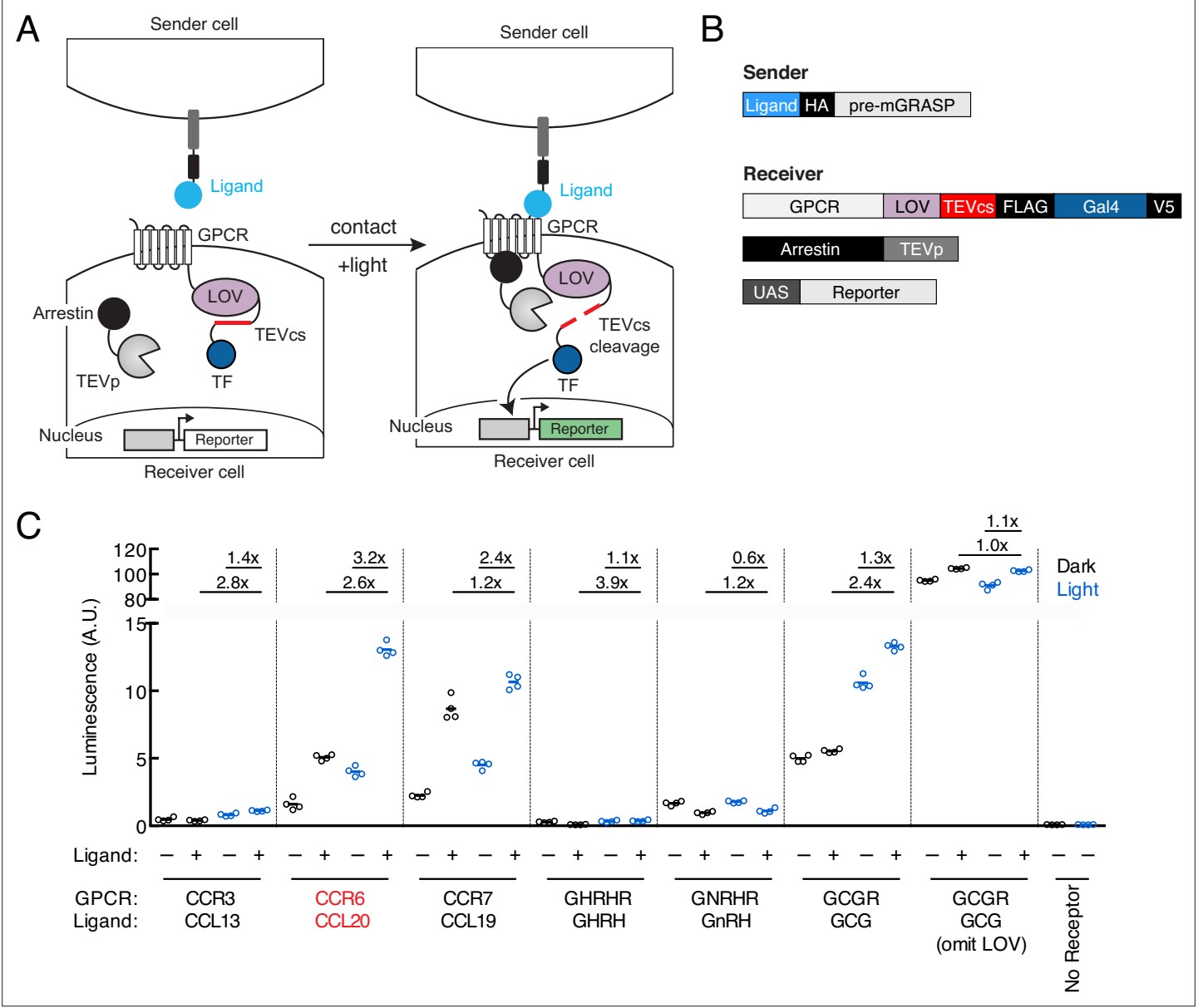

**Figure 1.** Design of TRACC (Transcriptional Readout Activated by Cell-cell Contacts). (**A**) Schematic of TRACC. A ligand is presented on sender cells; a GPCR specifically activated by the selected ligand is expressed in receiver cells. The GPCR is fused to the light-oxygen-voltage-sensing (LOV) domain, TEV protease cleavage site (TEVcs), and transcription factor (TF). Upon both cell-cell contact and exposure to blue light, the GPCR is activated and recruits arrestin fused to TEV protease (TEVp); blue light uncages the LOV domain, allowing cleavage of the TEVcs and subsequent release of the TF, which translocates to the nucleus and drives expression of a reporter of interest. (**B**) Constructs used in TRACC. The sender construct is comprised of a peptide ligand fused to pre-mGRASP (***Kim et al., 2011***) and the HA epitope tag. Receiver constructs include the corresponding GPCR fused to the LOV domain, TEVcs, and TF (Gal4), arrestin fused to TEVp, and a reporter construct. (**C**) Luciferase assay to screen a panel of GPCR-ligand pairs in cis. HEK293T cells were transfected with both sender and receiver constructs corresponding to each GPCR-ligand pair indicated, using the reporter UAS-luciferase. Approximately 8 hr after 10 min blue light exposure, the UAS-luciferase luminescence was recorded using a plate reader (n = 4 replicates per condition). The CCR6-CCL20 pair (red) showed the highest ±light and ±ligand signal ratios of 2.6-fold and 3.2-fold, respectively. A receiver construct using the glucagon receptor (GCGR), but omitting the LOV domain, analogous to that of previously published trans-Tango (***Talay et al., 2017***), was included as a control.

The online version of this article includes the following source data for figure 1:

**Source data 1.** Primary data for luminescence graphs in ***Figure 1C***.

amounts, incubation times, and light stimulation time (Methods). Approximately 8 hr following 10 min stimulation with blue light, TRACC activation was measured via a luciferase assay on a plate reader (*Figure 1C*). Of the GPCR-ligand pairs tested, the CCR6-CCL20 pair showed the highest ±light and ±ligand signal ratios (2.6- and 3.2-fold, respectively). Thus, we selected this GPCR-ligand pair for subsequent experiments. Of note, we included a construct that omitted the LOV domain, a design similar to trans-Tango (*Talay et al., 2017*), and observed high background reporter expression and ligand-independent activation (±ligand signal ratio of 1.1-fold), suggesting that the additional light-gate is crucial for minimizing background.

## Detecting cellular contacts in HEK293T cultures

Next, we tested whether TRACC could successfully detect cell-cell contacts in trans. To do this, sender and receiver HEK293T cells were separately transfected with the corresponding TRACC constructs (*Figure 2A*). Sender and receiver cell populations were co-plated together; approximately 8 hr following 10 min blue light stimulation, TRACC activation of UAS-luciferase expression was measured on a plate reader (*Figure 2B*). We observed a robust increase in luciferase reporter expression with ±light and ±sender signal ratios of 5.6-fold each. To demonstrate the versatility of a transcriptional reporter, we repeated the assay using a UAS-mCherry reporter in place of UAS-luciferase and performed immunostaining and confocal fluorescence imaging (*Figure 2C–D*). From the imaging assay, we observed robust light-dependent activation of the mCherry reporter in V5-positive (receiver-positive) cells that were in direct contact with HA-positive (sender-positive) cells. Quantitation of fluorescence intensities of cells across 50 fields of view (FOVs) showed that TRACC was highly specific; of 94 mCherry-expressing cells analyzed, 80.0% were in direct contact with an HA-positive sender cell (*Figure 2E*). While we did observe mCherry reporter activation in V5-positive cells not touching sender cells (20% of mCherry-expressing cells were not in direct contact with a sender cell), it is possible that these cells were previously in contact, but the sender cells were dislodged during the course of the experiment or during the washing steps in immunostaining. It is also possible that background activation may occur in cells expressing the arrestin-TEVp component at particularly high levels, which can result in GPCR activation-independent release of the TF (*Sanchez et al., 2020*). V5 intensity distributions were consistent across high-mCherry and low-mCherry populations (*Figure 2—figure supplement 1A*), suggesting that reporter activation is sender-dependent and not a result of differential receiver expression levels.

To assess sensitivity, we determined that 80.2% of receiver cells in contact with sender cells showed reporter expression above background (n = 106 cells from 50 FOVs; above background defined as having a fluorescence signal 1.5-fold or greater above a blank region). Furthermore, of the receiver cells in contact with an HA-positive (sender) cell, we also observed that the HA intensity within the same region of interest (ROI) was similar across both highly expressing and lowly expressing cells (*Figure 2—figure supplement 1B*), suggesting the difference in activation levels is not dependent on the expression levels of the sender construct. One possible explanation for contacting cells that do not turn on TF is that they may lack one of the other two receiver components (arrestin-TEVp or UAS-mCherry) that need to be co-transfected into the same cell for TRACC to function. We also performed this experiment using lentivirus transduction instead of transient transfection and similarly observed sender- and light-dependent reporter activation (*Figure 2—figure supplement 2*).

## Extending TRACC to neuronal systems

Recently developed non-viral tools for trans-synaptic tracing in neurons have expanded our ability to map synaptically connected cell populations. However, trans-Tango (*Talay et al., 2017*), TRACT (*Huang et al., 2017*), and BAcTrace (*Cachero et al., 2020*) have so far only been demonstrated in *Drosophila* and do not include mechanisms for temporal gating. To explore whether it would be feasible to adapt TRACC to neuronal systems, we cloned our constructs into AAV vectors driven by the synapsin promoter and utilized the orthogonal tTA-TRE TF system (*Figure 3A*). We generated mixed serotype AAV1/2 viruses for infecting cultured rat neurons. First, we verified that the individual constructs expressed and localized as expected in primary rat cortical neuron culture; we were able to detect the CCR6 and arrestin receiver components and observed that these constructs trafficked to neuronal processes as expected (*Figure 3—figure supplement 1A*). We also verified that the CCL20 sender construct localized properly to pre-synaptic terminals via colocalization with

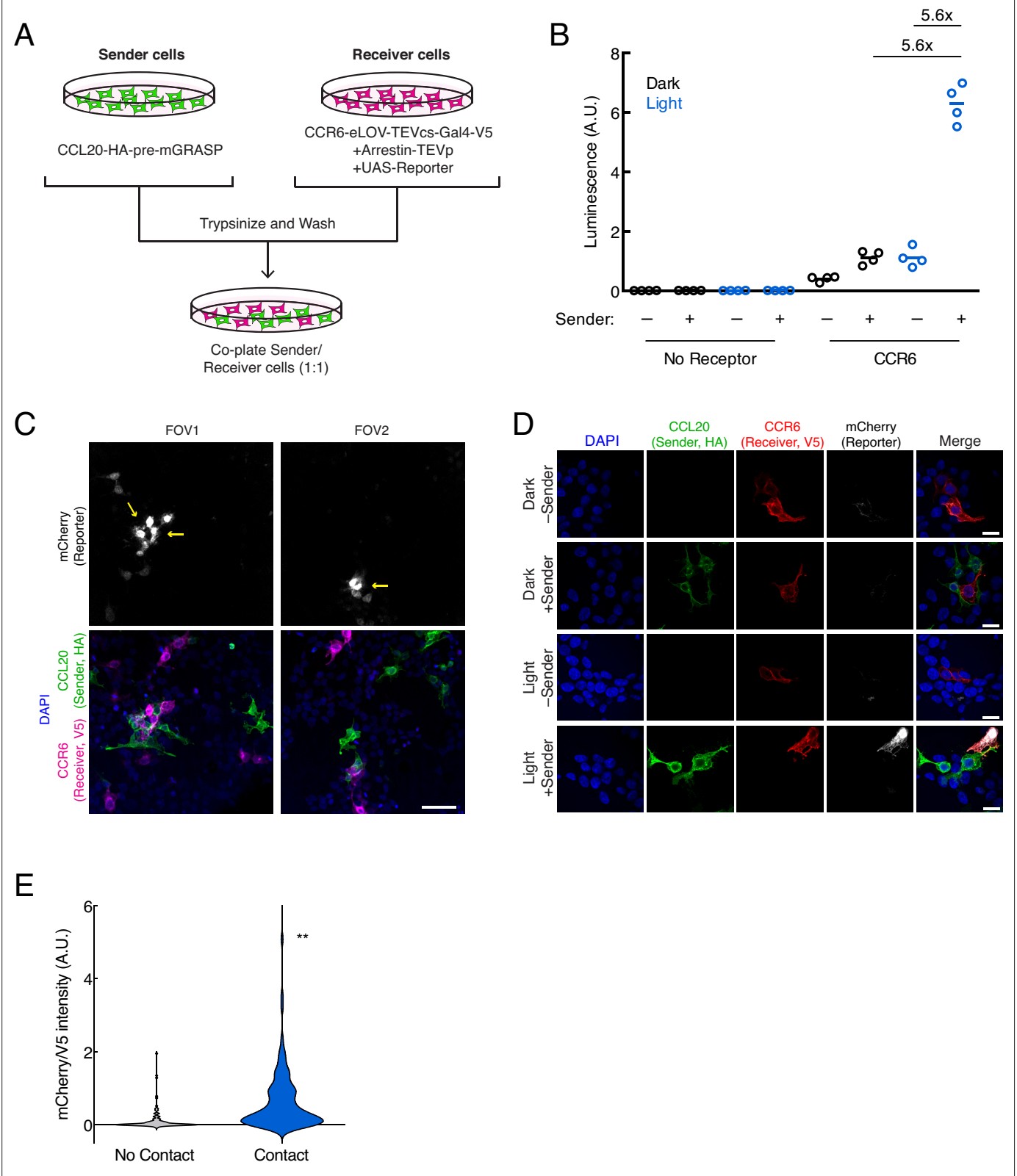

**Figure 2.** Using TRACC (Transcriptional Readout Activated by Cell-cell Contacts) to detect cell-cell contacts in HEK293T culture. (**A**) Experimental design for co-plating sender and receiver cells. (**B**) Luciferase assay using the CCR6-CCL20 GPCR-ligand pair in trans. HEK293T cells transfected with the CCL20 sender construct were co-plated with HEK293T cells transfected with receiver constructs and UAS-luciferase. Approximately 8 hr after 10 min blue light exposure, the UAS-luciferase luminescence was recorded (n = 4 replicates per condition). (**C**) Confocal fluorescence imaging of sender cells

*Figure 2 continued on next page*

*Figure 2 continued*

co-plated with receiver cells, using UAS-mCherry reporter. Approximately 8 hr after 10 min blue light exposure, cells were fixed and immunostained. mCherry activation occurs in receiver cells (V5-positive) that contact sender cells (HA-positive). Yellow arrowheads denote examples in which receiver cells are in contact with sender cells. Scale bar, 60 μm. (**D**) Confocal fluorescence imaging of sender cells co-plated with receiver cells, using UAS-mCherry at higher magnification. Cells were treated as in (**C**). Scale bars, 20 μm. (**E**) Quantification of mCherry/V5 intensity ratios for all V5-positive cells in the +light condition. The mCherry/V5 ratio was significantly higher in V5-positive cells that were in contact with HA-positive sender cells. (no contact, n = 108 cells; contact, n = 106 cells; two-tailed t-test, **p < 0.005).

The online version of this article includes the following source data and figure supplement(s) for figure 2:

**Source data 1.** Primary data for luminescence and cell count graphs in *Figure 2*.

**Figure supplement 1.** Additional quantification of HEK293T imaging in trans.

**Figure supplement 1—source data 1.** Primary data for graphs in *Figure 2—figure supplement 1*.

**Figure supplement 2.** Evaluation of TRACC (Transcriptional Readout Activated by Cell-cell Contacts) in HEK293T cells using lentiviral transduction.

---

endogenous synapsin (*Figure 3—figure supplement 1B-C*). Deletion of the intracellular neurexin domain disrupted its targeting specificity.

To test TRACC in neuron culture, we first co-expressed both receiver and sender constructs in the same population of neurons in cis and performed a luciferase assay. Approximately 24 hr after 10 min blue light stimulation, we measured luciferase reporter levels on a plate reader and observed robust activation of the TRE-luciferase reporter with high ±light and ±ligand signal ratios of 11.4- and 7.5-fold, respectively (*Figure 3B*). Next, we tested our tool in a co-culture system in which HEK293T cells expressing receiver constructs were co-plated onto neurons expressing a sender construct (*Figure 3C*). In the luciferase assay, we detected light- and sender-dependent gene expression, with ±light and ±sender signal ratios of 4.2- and 3.2-fold, respectively (*Figure 3D*). We repeated the assay with confocal microscopy imaging and again observed robust expression of the mCherry reporter only in the presence of both light and sender (*Figure 3E*). Lastly, we showed trans-cellular activation of TRACC in the reverse configuration, with sender HEK293T cells co-plated onto neurons expressing receiver constructs; both luciferase and imaging assays are shown in *Figure 3—figure supplement 1D-F*.

## Detecting interactions between neurons and glioma cells

High-grade gliomas are lethal brain cancers and the leading cause of brain tumor death in both children and adults (*Johung and Monje, 2017*). Recent studies have shown that neuronal interactions with glioma cells drive glioma progression (*Pan et al., 2021*; *Venkatesh et al., 2019*; *Venkatesh et al., 2017*; *Venkatesh et al., 2015*). Gliomas integrate into neural circuits, and one key mechanism driving glioma progression is signaling through functional neuron-to-glioma synapses (*Venkataramani et al., 2019*; *Venkatesh et al., 2019*). In these connections, pre-synaptic neurons communicate electrochemically to post-synaptic glioma cells, and the consequent inward current promotes glioma cell proliferation through membrane voltage-sensitive mechanisms (*Venkatesh et al., 2019*). How the synaptic connectivity evolves over the course of the cancer, which neurons form synapses with glioma cells, and which subpopulations of these cellularly heterogeneous tumors (*Filbin et al., 2018*; *Venteicher et al., 2017*) engage in neuron-to-glioma synapses has yet to be determined. We hypothesized that applying TRACC to experimental model systems of glioma may open the door to future studies of neuron-to-glioma connectivity at various timepoints in the evolution of the disease course as well as isolation of synaptic subpopulations for subsequent molecular analysis.

To see whether TRACC could be adapted for detecting contacts between neurons and glioma cells, we generated transposon-integrated cell lines stably expressing receiver constructs using patient-derived diffuse intrinsic pontine glioma (DIPG) cells (*Figure 4A*). We observed that SU-DIPG-VI cells stably expressing TRACC components exhibited low sensitivity; only a small fraction of cells showed mCherry reporter expression upon activation of TRACC with recombinant ligand and exogenous blue light (*Figure 4A*; *Figure 4—figure supplement 1A*). To further optimize TRACC in this system, we generated five additional receiver cell lines containing variants of either the LOV domain or the TEVp. We compared eLOV versus hLOV, which combines features of eLOV and iLiD (*Kim et al., 2017*). We also compared wild-type TEVp with faster variants that were engineered via directed evolution, uTEV1 and uTEV2 (*Sanchez and Ting, 2020*). From screening the SU-DIPG-VI cell lines expressing the

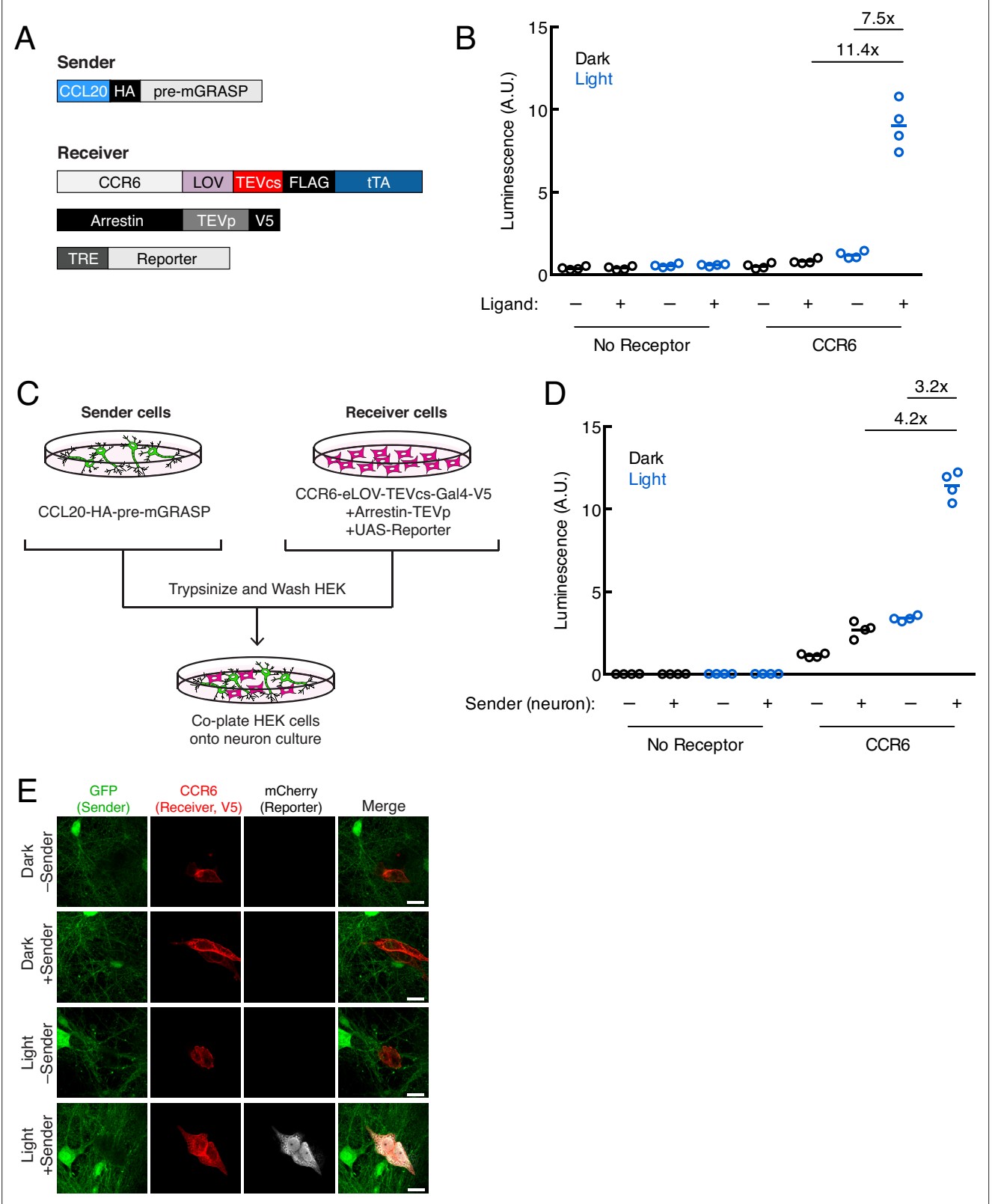

**Figure 3.** Using TRACC (Transcriptional Readout Activated by Cell-cell Contacts) to detect contacts in neuron culture and HEK293T-neuron co-culture. (**A**) Constructs used in TRACC in neuron culture. CCR6 is the GPCR and CCL20 is its activating peptide ligand. For expression in neurons, the transcription factor (TF) is changed from Gal4 to tTA and the reporter gene is driven by TRE rather than a UAS promoter. (**B**) Luciferase assay using TRACC constructs expressed in cis in neuron culture. Primary rat cortical neurons were infected with AAV1/2 viruses encoding both sender and receiver

*Figure 3 continued on next page*

*Figure 3 continued*

constructs, including the reporter TRE-luciferase, on DIV5 and light-stimulated on DIV10. Approximately 24 hr after 10 min blue light exposure, the TRE-luciferase luminescence was recorded using a plate reader (n = 4 replicates per condition). (**C**) Experimental design for co-plating sender neurons and receiver HEK293T cells. (**D**) Luciferase assay using sender neurons co-cultured with receiver HEK293T cells. Primary rat cortical neurons were infected with AAV1/2 viruses encoding the sender construct on DIV5. HEK293T cells expressing receiver constructs and UAS-luciferase were co-plated onto sender neurons on DIV9, and the resulting co-culture was light-stimulated on DIV10. Approximately 8 hr after 10 min blue light exposure, the UAS-luciferase luminescence was recorded (n = 4 replicates per condition). (**E**) Confocal fluorescence imaging of receiver HEK293T cells co-cultured with sender neurons, using UAS-mCherry. GFP driven by the synapsin promoter was included as an infection marker to visualize transduced neurons. Cells were treated as in (**D**). Scale bars, 20 μm.

The online version of this article includes the following source data and figure supplement(s) for figure 3:

**Source data 1.** Primary data for luminescence graphs in *Figure 3*.

**Figure supplement 1.** Additional characterization of TRACC (Transcriptional Readout Activated by Cell-cell Contacts) constructs in neuron culture.

**Figure supplement 1—source data 1.** Primary data for colocalization and luminescence graphs in *Figure 3—figure supplement 1*.

different combinations of LOV and TEVp, we found that eLOV in combination with uTEV2 showed the highest sensitivity (*Figure 4A*; *Figure 4—figure supplement 1A*). Furthermore, while the proportion of cells that became activated increased, this activation was still highly specific and required the presence of both recombinant ligand and blue light (*Figure 4A*). We tested this cell line in a co-culture system in which SU-DIPG-VI cells stably expressing receiver constructs (with eLOV and uTEV2) were co-plated onto neurons expressing the sender construct (*Figure 4B*). In this assay, we observed robust activation of the mCherry reporter only in the presence of both light and sender expression in sender neurons (*Figure 4C*; *Figure 4—figure supplement 1B*), demonstrating the potential to map neuron-glioma cell-cell contact interactions in patient-derived glioma model systems.

## Discussion

We have adapted our previously published SPARK tool (*Kim et al., 2017*), an assay for detecting protein-protein interactions, into a tool for detecting cell-cell contacts. TRACC is a GPCR-based detector of cell-cell interactions with transcriptional readout, offering versatile outputs for detection and downstream manipulation. TRACC incorporates the light-sensitive LOV domain, such that tool activation can only occur in a user-defined window during which exogenous blue light is supplied. Compared to tools that directly label contact sites such as LIPSTIC (*Pasqual et al., 2018*), split-HRP (*Martell et al., 2016*), and mGRASP (*Kim et al., 2011*), TRACC provides genetic access to contacting cell populations for downstream analysis and potential manipulation. Furthermore, in comparison to other TF-based tools like synNotch (*Morsut et al., 2016*) and trans-Tango (*Talay et al., 2017*), the incorporation of light gating in TRACC substantially reduces background signal and provides temporal specificity for detecting cell-cell contacts during user-defined time windows.

We used TRACC to detect cell-cell contacts between separately transfected HEK293T cell populations and observed specific and sensitive tool activation. We further showed that TRACC can be applied to detect cellular contacts in both neuron and glioma systems, and that the sender construct localizes properly to pre-synaptic terminals. In future studies, TRACC may be useful for synapse-specific tracing, particularly in the anterograde direction (pre-synaptic to post-synaptic) for which tools are currently lacking. This will first require careful validation of TRACC component localization in vivo and testing of TRACC specificity and sensitivity in a well-characterized circuit in vivo. TRACC may also be useful for future investigations of neuron-glioma circuitry, allowing identification and subsequent analysis of connected subpopulations.

## Materials and methods

Table of plasmids used in this study.

| Plasmid name | Plasmid vector | Promoter | Features | Variants | Details | Used for | Used in |
|---|---|---|---|---|---|---|---|
| P1-P6 | pAAV | CMV | HindIII-GPCR-SpeI-NES-NheI-eLOV-TEVcs-FLAG-Gal4-V5 | GPCRs: CCR3, CCR6, CCR7, GHRHR, GNRHR, GCGR | NES: ELAEKLAGLDIN; TEVcs: ENLYFQM; FLAG: DYKDDDDK; V5: GKPIPNPLLGLDST | Transient expression | *Figures 1–3*; *Figure 2—figure supplement 1* |
| P7 | pAAV | CMV | HindIII-GPCR-SpeI-NES-NheI- TEVcs-FLAG-Gal4-V5 | GPCR: GCGR | NES: ELAEKLAGLDIN; TEVcs: ENLYFQM; FLAG: DYKDDDDK; V5: GKPIPNPLLGLDST | Transient expression | *Figure 1* |
| P8-13 | pCAG | CAG | KpnI-Ligand-18 aa linker-AgeI-3xHA-AgeI-pre-mGRASP | Ligands: CCL13, CCL20, CCL19, GHRH, GnRH, GCG | 18 aa linker: GNGNGNGNGNGNGNGNGN; 3xHA: AAVYPYDVPDYAGYPYDVPDYAGSYPYDVPDYAPAA | Transient expression | *Figures 1 and 2*; *Figure 2—figure supplements 1 and 2* |
| P14 | pCDNA3 | CMV | BsaI-myc-Arrestin-10 aa linker-TEVp | | 10 aa linker: GGSGSGSGGS | Transient expression | *Figures 1–3*; *Figure 2—figure supplement 1* |
| P15-16 | pAAV | UAS | Reporter | Reporters: Luciferase, mCherry | | Transient expression | *Figures 1–3*; *Figure 2—figure supplement 1* |
| P17 | AAV1 | | | | | Producing AAV1/2 virus | *Figures 3 and 4*; *Figure 3—figure supplement 1* |
| P18 | AAV2 | | | | | Producing AAV1/2 virus | *Figures 3 and 4*; *Figure 3—figure supplement 1* |
| P19 | DF6 | | | | | Producing AAV1/2 virus | *Figures 3 and 4*; *Figure 3—figure supplement 1* |
| P20 | pAAV | Synapsin | BamHI-CCL20-18 aa linker-AgeI-3xHA-AgeI-pre-mGRASP | | 18 aa linker: GNGNGNGNGNGNGNGNGN; 3xHA: AAVYPYDVPDYAGYPYDVPDYAGSYPYDVPDYAPAA | AAV-induced expression in neurons | *Figures 3 and 4*; *Figure 3—figure supplement 1* |
| P21 | pAAV | Synapsin | BamHI- CCR6-SpeI-NES-NheI-eLOV-TEVcs-FLAG-tTA | | NES: ELAEKLAGLDIN; TEVcs: ENLYFQM; FLAG: DYKDDDDK | AAV-induced expression in neurons | *Figure 3*; *Figure 3—figure supplement 1* |
| P22 | pAAV | Synapsin | Arrestin-10 aa linker-TEVp-V5 | | 10 aa linker: GGSGSGSGGS; V5: GKPIPNPLLGLDST | AAV-induced expression in neurons | *Figure 3*; *Figure 3—figure supplement 1* |
| P23-24 | pAAV | TRE | Reporter | Reporters: Luciferase, mCherry | | AAV-induced expression in neurons | *Figure 3*; *Figure 3—figure supplement 1* |
| P25 | pAAV | Synapsin | GFP | | | AAV-induced expression in neurons | *Figure 3*, 4 |
| P26-27 | pPB | EF-1α | AgeI-CCR6- SpeI-NES-NheI-LOV-TEVcs-FLAG-Gal4 | LOV variants: eLOV, hLOV | NES: ELAEKLAGLDIN; TEVcs: ENLYFQM; FLAG: DYKDDDDK | Stable expression in DIPG | *Figure 4*; *Figure 4—figure supplement 1* |
| P28-30 | pPB | UbC; UAS | Arrestin-10 aa linker-TEVp-V5; UAS-mCherry | TEVp variants: WT TEVp, uTEV1, uTEV2 | 10 aa linker: GGSGSGSGGS; V5: GKPIPNPLLGLDST | Stable expression in DIPG | *Figure 4*; *Figure 4—figure supplement 1* |
| P31 | pPB | | | | Super PiggyBac Transposase Expression Vector (System Biosciences) | Stable expression in DIPG | *Figure 4*; *Figure 4—figure supplement 1* |
| P31 | pCMV | CMV | dR8.91 | | | Producing lentivirus | *Figure 2—figure supplement 2* |
| P32 | pCMV | CMV | VSV-G | | | Producing lentivirus | *Figure 2—figure supplement 2* |
| P33 | pLX208 | CMV | CCR6- SpeI-NES-NheI-eLOV-TEVcs-FLAG-Gal4 | | NES: ELAEKLAGLDIN; TEVcs: ENLYFQM; FLAG: DYKDDDDK | Lentivirus-induced expression in HEK293T | *Figure 2—figure supplement 2* |
| P34 | pLX208 | CMV | Arrestin-10 aa linker-TEVp | | 10 aa linker: GGSGSGSGGS | Lentivirus-induced expression in HEK293T | *Figure 2—figure supplement 2* |

| Plasmid name | Plasmid vector | Promoter | Features | Variants | Details | Used for | Used in |
|---|---|---|---|---|---|---|---|
| P35 | pLX208 | CMV | CCL20-18 aa linker-AgeI-3xHA-AgeI-pre-mGRASP | | 18 aa linker: GNGNGNGNGNGNGNGNGN; 3xHA: AAVYPYDVPDYAGYPYDVPD YAGSYPYDVPDYAPAA | Lentivirus-induced expression in HEK293T | *Figure 2—figure supplement 2* |
| P36 | pLX208 | UAS | mCherry | | | Lentivirus-induced expression in HEK293T | *Figure 2—figure supplement 2* |

Table of antibodies used in this study.

| Antibody | Source | Vendor | Catalog number | Dilution(s) |
|---|---|---|---|---|
| Anti-V5 | Mouse | Thermo Fisher Scientific | R96025 | WB: 1:10,000; IF: 1:1000 |
| Anti-HA | Rabbit | Cell Signaling Technology | C29F4 | WB: 1:5000; IF: 1:1000 |
| DAPI | - | Enzo Life Sciences | AP402-0010 | IF: 1 µg/mL final concentration |
| Anti-mouse-AlexaFluor488 | Goat | Thermo Fisher Scientific | A11029 | IF: 1:1000 |
| Anti-mouse-AlexaFluor568 | Goat | Thermo Fisher Scientific | A11031 | IF: 1:1000 |
| Anti-mouse-AlexaFluor647 | Goat | Thermo Fisher Scientific | A21236 | IF: 1:1000 |
| Anti-rabbit-AlexaFluor488 | Goat | Thermo Fisher Scientific | A11008 | IF: 1:1000 |
| Anti-rabbit-AlexaFluor568 | Goat | Thermo Fisher Scientific | A11036 | IF: 1:1000 |
| Anti-rabbit-AlexaFluor647 | Goat | Thermo Fisher Scientific | A21245 | IF: 1:1000 |
| Anti-RFP | Rabbit | Rockland | 600-401-379 | IF: 1:1000 |
| Anti-VP16 | Rabbit | Abcam | Ab4808 | IF: 1:1000 |
| Anti-Synapsin | Guinea Pig | Synaptic Systems | 106 004 | IF: 1:500 |
| Anti-NFH | Chicken | Aves Labs | NFH | IF: 1:2000 |
| RFP-Booster AlexaFluor 568 | Alpaca | Chromotek | rb2AF568-50 | IF: 1:500 |

## Cloning

All constructs were generated using standard cloning techniques. PCR fragments were amplified using Q5 polymerase (NEB). Vectors were digested using enzymatic restriction digest and ligated to gel purified PCR products using Gibson assembly. Ligated plasmid products were transformed into either competent XL1-Blue bacteria or competent NEB Stable bacteria (C3040H).

## Cell lines

HEK293T cells were obtained from ATCC; SU-DIPG cell lines were patient derived. All cell lines have been tested mycoplasma negative.

## HEK293T cell culture and transfection

HEK293T cells from ATCC were cultured as a monolayer in complete media: Dulbecco's modified Eagle's medium (DMEM) with 4.5 g/L glucose and L-glutamine supplemented with 10% (w/v) fetal bovine serum, 1% (v/v) GlutaMAX, 50 units/mL penicillin, and 50 µg/mL streptomycin at 37°C under 5% $CO_2$. For confocal imaging experiments, glass coverslips were coated with 50 µg/mL fibronectin in DPBS for at least 20 min at room temperature before plating; cells were grown on glass coverslips in 24-well plates with 500 µL growth medium. For luciferase assays, cells were grown in 24-well plates with 500 µL growth medium, transfected, and replated into 96-well plates with 100 µL growth medium (20,000 cells/well) 4 hr after transfection.

For transient expression, cells were transfected at approximately 70% confluency using 5 µL/mL Lipofectamine2000 and corresponding expression plasmids in serum-free media. Complete

transfection protocols for individual experiments are described below. Note that for all HEK293T cell experiments expressing receiver constructs, it is critical to light-stimulate cultures 15–18 hr post-transfection and to perform the assay (luciferase or fixation for imaging) 7–9 hr post-stimulation to avoid background accumulation.

## Primary rat cortical neuron culture and AAV infection

Cortical neurons were harvested from rat embryos euthanized at embryonic day 18 and plated in 24-well plates or 48-well plates as previously described (*Loh et al., 2016*; *Wang et al., 2017*). Plates were coated with 0.001% (w/v) poly-L-ornithine in DPBS at room temperature overnight, washed twice with DPBS, and subsequently coated with 5 µg/mL of mouse laminin in DPBS at 37°C for at least 4 hr. Neurons were cultured in complete neuronal media: 1:1 Advanced DMEM:neurobasal, supplemented with 2% (v/v) B27 supplement, 5% (v/v) fetal bovine serum, 1% GlutaMAX, 50 units/mL penicillin, 50 µg/mL streptomycin, 0.1% (v/v) β-mercaptoethanol, 5 ng/mL glial-derived neurotrophic factor, and 5 µM TRO19622 at 37°C under 5% $CO_2$. At DIV2, half of the media was removed from each well and replaced with complete neuronal media supplemented with 10 µM 5-fluorodeoxyuridine to inhibit glial cell growth. Half of the media was replaced with complete neuronal media every 2 days afterwards.

## SU-DIPG-VI culture and transposon integration

DIPG cells (SU-DIPG, an H3.3K27M + patient-derived neurosphere culture) were cultured in tumor stem media: neurobasal (-A), supplemented with 20 ng/mL human bFGF, 20 ng/mL human EGF, 10 ng/mL human PDGF-AA, 10 ng/mL PDGF-BB, and 2 ng/mL heparin at 37°C under 5% $CO_2$.

To generate transposon-integrated stable cell lines, naive glioma cells were plated in tumor stem media in six-well plates. Plates were coated first with 0.01% poly-D-lysine for 20 min and then with 5 µg/mL of mouse laminin for 3 hr. Approximately 400,000 cells were plated per well, and cells were incubated overnight. Receiver constructs were introduced sequentially. Once cells were fully adhered, 1.5 µg of a receiver construct was added to 8.5 µL of FuGene HD (Promega), 0.8 µg of Super PiggyBac Transposase Expression Vector, and 30 µL of OptiMEM serum-free media per condition. Approximately 5 hr after transfection, a half-media change was performed to remove FuGene HD toxicity. Adherent glioma cells were expanded into 10 cm² Petri dishes, after which antibiotic selection was initiated (1 µg/mL blasticidin or 100 µg/mL G418 [Geneticin]). The transfection protocol was repeated to introduce the second receiver construct. Cells were maintained under double selection conditions until needed for in vitro experiments.

## Sample-size estimation and replication

No statistical methods were used to determine sample size, and instead relied on guidelines from previously published works. For luminescence assays, we used at least four technical replicates. Sample sizes are listed in figure legends. All experiments were replicated at least once (biological replicates). Replicates are listed in figure legends.

## Luciferase assays with HEK293T

For experiments with a luciferase reporter, HEK293T cells were cultured in 24-well plates and transfected with 70 ng of pAAV-CMV-GPCR-eLOV-TEVcs-Gal4, 20 ng of pAAV-CMV-Arrestin-TEVp, and 30 ng of pAAV-UAS-luciferase. For conditions with cis activation, 50 ng of pCAG-CAG-Sender-pre-mGRASP was also included. For each condition, plasmid DNA was mixed with 2.5 µL Lipofectamine2000 in 50 µL serum-free DMEM and incubated at room temperature for 20 min. The DNA-Lipofectamine2000 mix was then added directly to each well. Cells were then incubated for 4 hr in a 37°C incubator. Cells were then lifted using 100 µL Trypsin and resuspended in complete media and pelleted by centrifugation for 3 min at 200 g. Cells were then replated into 96-well white, clear-bottom microplates at a density of 20,000 cells/well. Plates were wrapped in aluminum foil and incubated for an additional 12 hr in a 37°C incubator.

For light stimulation, cells were exposed to an LED light array (467 nm, 60 mW/cm², 1 s of light every 3 s) at 37°C for 10 min. After light stimulation, the plate was rewrapped in aluminum foil and incubated for an additional 8 hr in a 37°C incubator.

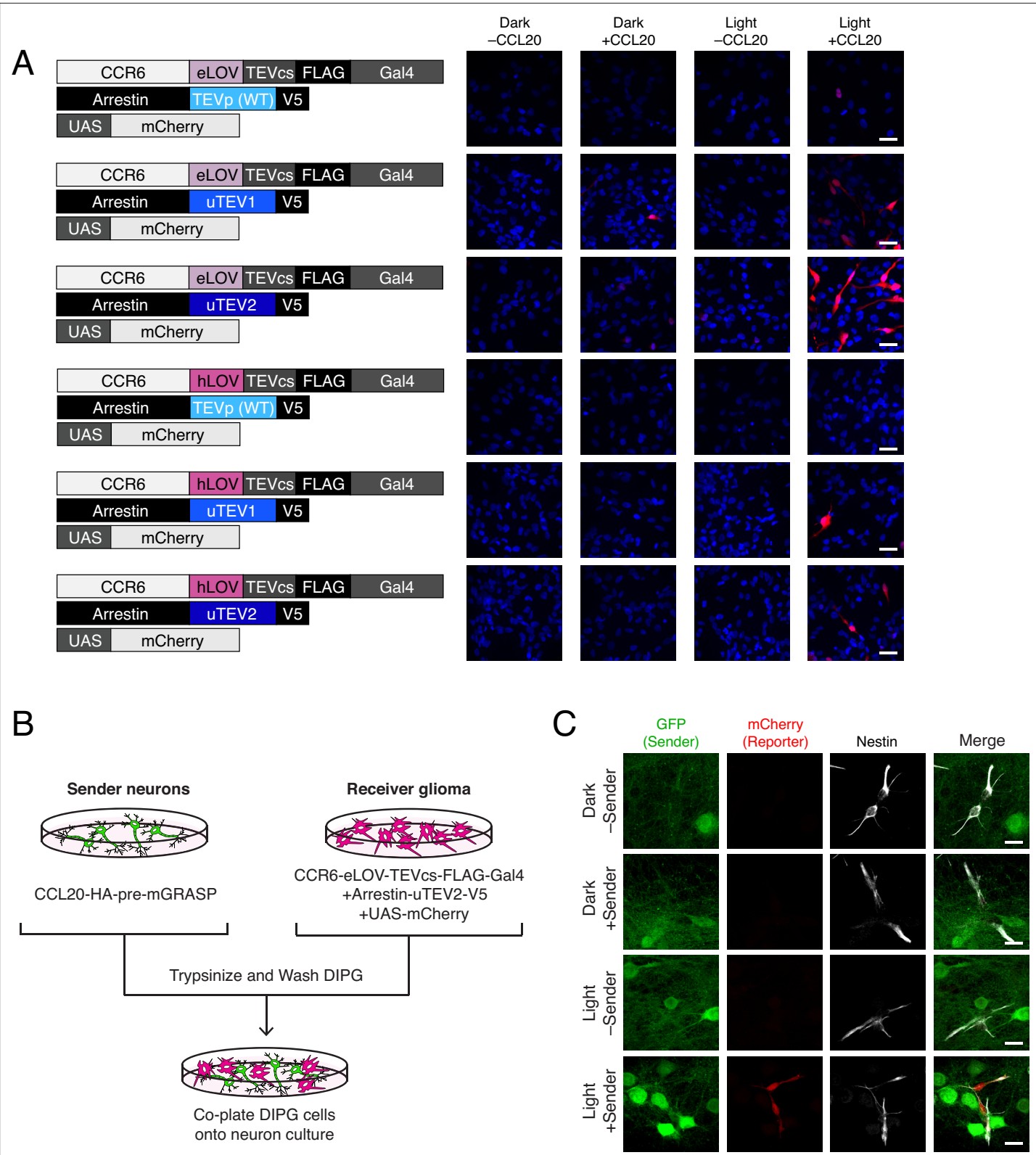

**Figure 4.** Using TRACC (Transcriptional Readout Activated by Cell-cell Contacts) to detect contacts in DIPG (diffuse intrinsic pontine glioma) culture. (**A**) Optimization of TRACC components in transposon-integrated SU-DIPG-VI stable cell lines. We compared TRACC constructs containing eLOV or hLOV, and WT TEVp, uTEV1, or uTEV2. Cells were plated and treated with 0.2 μg/mL recombinant CCL20 and 10 min blue light. Approximately 24 hr after blue light exposure, cells were fixed and immunostained. Scale bar, 30 μm. (**B**) Experimental design for co-plating sender neurons and receiver DIPG cells. (**C**)

*Figure 4 continued on next page*

Figure 4 continued

Confocal fluorescence imaging of DIPG glioma expressing receiver constructs containing eLOV and uTEV2 co-plated with sender neurons. Primary rat cortical neurons were infected with AAV1/2 viruses encoding the sender construct on DIV5. DIPG cells were co-plated onto sender neurons on DIV9, and the resulting co-culture was light-stimulated on DIV11. Approximately 24 hr after 10 min blue light exposure, cells were fixed and immunostained. GFP driven by the synapsin promoter was included as an infection marker to visualize transduced neurons; Nestin is a marker for DIPG cells. Scale bar, 20 µm.

The online version of this article includes the following source data and figure supplement(s) for figure 4:

**Figure supplement 1.** Quantification of activation in diffuse intrinsic pontine glioma (DIPG) stable lines.

**Figure supplement 1—source data 1.** Primary data for cell count graphs in *Figure 4—figure supplement 1*.

For luciferase reporter measurements, the Bright-Glo Luciferase Assay System (Promega) was used. The Bright-Glo reagent was thawed at room temperature 1 hr prior to use. Media was aspirated from each well, and each well was then washed with 100 µL DPBS. Next, 50 µL DPBS and 50 mL Bright-Glo reagent were added to each well. Luminescence was analyzed 3 min later at 25°C on a plate reader (Tecan Infinite M1000 Pro) using a 1000 ms acquisition time, the Green-1 filter, and linear shaking for 3 s.

## HEK293T co-culture for trans assays

For trans assays using HEK293T cells, cells were cultured in 24-well plates as described above. Receiver cells were transfected with 70 ng of pAAV-CMV-CCR6-eLOV-TEVcs-Gal4, 20 ng of pAAV-CMV-Arrestin-TEVp, and 30 ng of pAAV-UAS-luciferase. Sender cells were transfected with 50 ng of pCAG-CAG-CCL20-pre-mGRASP. For each condition, plasmid DNA was mixed with 2.5 µL Lipofectamine2000 in 50 µL serum-free DMEM and incubated at room temperature for 20 min. The DNA-Lipofectamine2000 mix was then added directly to each well. Cells were then incubated for 4 hr in a 37°C incubator. Cells were then lifted using 100 µL Trypsin and resuspended in complete media and pelleted by centrifugation for 3 min at 200 g, and further washed with DPBS twice. Receiver and sender cells were mixed at a 1:1 ratio and cells were then replated into 96-well white, clear-bottom microplates at a density of 20,000 cells/well. Plates were wrapped in aluminum foil and incubated for an additional 12 hr in a 37°C incubator. The luciferase reporter assay was performed as described above.

## Lentivirus generation and HEK293T co-culture for trans assays via lentivirus

To generate lentivirus, HEK293T cells were cultured in T75 flasks and transfected at approximately 70% confluency with 7500 ng of the lentiviral vector of interest and packaging plasmids pCMV-dR8.91 (6750 ng) and pCMV-VSV-G (750 ng) with 75 µL of polyethyleneimine (PEI, 1 mg/mL; Polysciences). Approximately 72 hr after transfection, the cell medium was collected, centrifuged for 3 min at 300 g to remove cell debris, and filtered through a 0.45 µm filter. Filtered media containing lentivirus was then centrifuged at 3000 g in 100 K Millipore Amicon Filters until concentrated approximately four-fold, and then aliquoted into 0.5 mL aliquots, flash-frozen in liquid nitrogen, and stored at –80°C. Prior to infection, viral aliquots were thawed at 37°C.

For lentivirus trans assays in HEK293T cells, cells were cultured in 12-well plates coated with 50 µg/mL fibronectin in DPBS for at least 20 min at room temperature before plating. Cells were infected with lentivirus at approximately 30% confluency. Receiver cells were infected with 100 µL of concentrated pLX208-CMV-CCR6-eLOV-TEVcs-Gal4, 50 µL of pLX208-CMV-Arrestin-TEVp, and 50 µL of pLX208-UAS-mCherry. Sender cells were infected with 100 µL of pLX208-CMV-CCL20-pre-mGRASP. Cells were wrapped in aluminum foil to protect them from light and incubated for 48 hr in a 37°C incubator. After 48 hr, cells were lifted and co-plated under red light to not expose cells to white/blue light. Cells were lifted using 500 µL Trypsin, resuspended in complete media, and pelleted by centrifugation for 3 min at 300 g. Receiver and sender cells were resuspended in complete media and mixed at a 1:1 ratio and cells were then replated onto fibronectin-coated glass coverslips in 24-well plates at a density of 200,000 cells/well. Plates were wrapped in aluminum foil and incubated for an additional 18 hr in a 37°C incubator. The mCherry reporter assay was performed as described above, with the additional use of RFP-Booster AlexaFluor 568 (Chromotek) for amplifying mCherry signal.

## AAV1/2 production in HEK293T

To generate supernatant AAV, HEK293T cells were cultured in T25 flasks and transfected at approximately 70% confluency with 900 ng of the AAV vector containing the gene of interest and AAV packaging/helper plasmids AAV1 (450 ng), AAV2 (450 ng), and DF6 (1800 ng) with 25 µL PEI in water (pH 7.3, 1 mg/mL) in serum-free media. After 48 hr, the cell medium containing the AAV was harvested and filtered using a 0.45 µm filter.

## Luciferase assays with neuron culture

Primary rat cortical neurons were harvested and cultured in 48-well plates as described above. At DIV5, supernatant AAV1/2 generated as described above were added to each well as follows. For expression of receiver constructs, 50 µL of AAV1/2 encoding pAAV-Syn-CCR6-eLOV-TEVcs-tTA, 20 µL of AAV1/2 encoding pAAV-Syn-Arrestin-TEVp, and 20 µL of AAV1/2 encoding pAAV-TRE-luciferase were added directly to the well. For conditions with cis activation, 50 µL of AAV1/2 encoding pAAV-Syn-CCL20-pre-mGRASP was also added to the well. After the media change on DIV8, plates were wrapped in aluminum foil. At DIV10, cells were exposed to an LED light array (467 nm, 60 mW/cm$^2$, 1 s of light every 3 s) at 37°C for 10 min. After light stimulation, the plate was rewrapped in aluminum foil and incubated for an additional 24 hr in a 37°C incubator.

For luciferase reporter measurements, the Bright-Glo Luciferase Assay System (Promega) was used. The Bright-Glo reagent was thawed at room temperature 1 hr prior to use. Media was aspirated from each well, and each well was then washed with 200 µL DPBS. Next, 50 µL DPBS and 50 mL Bright-Glo reagent were added to each well, and the resulting lysates were transferred to 96-well white, clear-bottom microplates. Luminescence was analyzed at 25°C on a plate reader (Tecan Infinite M1000 Pro) using a 1000 ms acquisition time, the Green-1 filter, and linear shaking for 3 s.

## Neuron and HEK293T co-culture

Primary rat cortical neurons and HEK293T cells were cultured as described above. For co-culture assays with sender neurons and receiver HEK293T cells, 50 µL of AAV1/2 encoding pAAV-Syn-CCL20-pre-mGRASP was added to neurons at DIV5. At DIV9, HEK293T cells were separately transfected with 70 ng of pAAV-CMV-CCR6-eLOV-TEVcs-Gal4, 20 ng of pAAV-CMV-Arrestin-TEVp, and 30 ng of pAAV-UAS-luciferase, as described above. After 4 hr, HEK293T cells were then lifted using 100 µL Trypsin and resuspended in complete media and pelleted by centrifugation for 3 min at 200 g, and further washed with DPBS twice. HEK293T cells were then resuspended in complete neuronal media and added directly to the neuron culture at a density of 5000 cells/well for a 48-well plate or 10,000 cells/well for a 24-well plate; plates were then wrapped in aluminum foil. At DIV10, cells were exposed to an LED light array (467 nm, 60 mW/cm$^2$, 1 s of light every 3 s) at 37°C for 10 min. After light stimulation, the plate was rewrapped in aluminum foil and incubated for an additional 8 hr in a 37°C incubator. The luciferase reporter assay was performed as described above.

For co-culture assays with receiver neurons and sender HEK293T cells, 50 µL of AAV1/2 encoding pAAV-Syn-CCR6-eLOV-TEVcs-tTA, 20 µL of AAV1/2 encoding pAAV-Syn-Arrestin-TEVp, and 20 µL of AAV1/2 encoding pAAV-TRE-luciferase were added to neurons at DIV5. At DIV9, HEK293T cells were separately transfected with 50 ng of pCAG-CAG-CCL20-pre-mGRASP, as described above. After 4 hr, HEK293T cells were then lifted using 100 µL Trypsin and resuspended in complete media and pelleted by centrifugation for 3 min at 200 g, and further washed with DPBS twice. HEK293T cells were then resuspended in complete neuronal media and added directly to the neuron culture at a density of 5000 cells/well for a 48-well plate or 10,000 cells/well for a 24-well plate; plates were then wrapped in aluminum foil. At DIV10, cells were exposed to an LED light array (467 nm, 60 mW/cm$^2$, 1 s of light every 3 s) at 37°C for 10 min. After light stimulation, the plate was rewrapped in aluminum foil and incubated for an additional 24 hr in a 37°C incubator. The luciferase reporter assay was performed as described above.

## Cell culture fixation, staining, and confocal imaging

For immunofluorescence experiments, cells were cultured, transfected, infected, and/or co-plated as described above. pAAV-UAS-mCherry and pAAV-TRE-mCherry were used in place of pAAV-UAS-luciferase and pAAV-TRE-luciferase. After incubation for the indicated times post-stimulation, cell cultures or co-cultures were fixed with 4% (v/v) paraformaldehyde diluted in serum-free media and

20% (v/v) 5× PHEM buffer (300 mM PIPES, 125 mM HEPES, 50 mM EGTA, 10 mM MgCl$_2$, 0.6 M sucrose, pH 7.3) for 10 min. The solution was removed and cells were then permeabilized with cold methanol at 4°C for 10 min. Cells were then washed three times with PBS and blocked in 1% BSA (w/v) in PBS for 30 min at room temperature. Cells were incubated with primary antibody in 1% BSA (w/v) in PBS for 3 hr at room temperature. After washing three times with PBS, cells were incubated with DAPI, secondary antibodies, and neutravidin-AlexaFluor 647 in 1% BSA (w/v) in PBS for 1 hr at room temperature. Cells were then washed with PBS three times, mounted onto glass slides, and imaged by confocal fluorescence microscopy. Confocal imaging was performed with a Zeiss AxioObserver inverted microscope with a 20× air objective, and 40× and 63× oil-immersion objectives. The following combinations of laser excitation and emission filters were used for various fluorophores: DAPI (405 nm laser excitation, 445/40 nm emission), AlexaFluor 488 (491 nm laser excitation, 528/38 nm emission), AlexaFluor 568 (561 nm laser excitation, 617/73 nm emission), and AlexaFluor 647 (647 nm laser excitation, 680/30 nm emission). All images were collected with SlideBook (Intelligent Imaging Innovations) and processed with ImageJ.

## Neuron and DIPG co-culture

Primary rat cortical neurons and SU-DIPG-VI cells were cultured as described above. For co-culture assays with sender neurons and receiver SU-DIPG-VI cells, 50 µL of AAV1/2 encoding pAAV-Syn-CCL20-pre-mGRASP was added to neurons at DIV5. For subsequent media changes after neuron infection with AAV, complete neuronal media without serum was used instead. At DIV9, stable DIPG cells expressing receiver constructs were dissociated using TrypLE and then pelleted by centrifugation for 3 min at 200 g, and further washed with DPBS twice. DIPG cells were then resuspended in complete neuronal media without serum and added directly to the neuron culture at a density of 10,000 cells/well for a 24-well plate; plates were then wrapped in aluminum foil. At DIV11, cells were exposed to an LED light array (467 nm, 60 mW/cm$^2$, 1 s of light every 3 s) at 37°C for 10 min. After light stimulation, the plate was rewrapped in aluminum foil and incubated for an additional 24 hr in a 37°C incubator. Cells were then fixed and immunostained as described above.

## Specificity and sensitivity analysis for HEK293T trans assay

For specificity analysis of imaging data in *Figure 2C-E*, an ROI was generated for each V5-positive cell (receiver). The average pixel intensities in each channel for each manually drawn ROI were measured. Pixel intensities were corrected for background by subtracting the average pixel intensities of 50 V5-, mCherry-, and HA-negative cells for each channel. Measurements for each ROI were separated into either V5-positive cells in contact (n = 106 cells) with an HA-positive (sender) cell or not in contact (n = 108 cells), and the mCherry/V5 signal ratios were plotted (*Figure 2E*).

For sensitivity analysis of imaging data in *Figure 2C–E*, the mCherry signal in V5-positive (receiver) cells in contact with an HA-positive (sender) cell were measured as described above. Cells were considered mCherry-positive if the signal was greater than 1.5-fold over background, as determined from V5-negative cells. From this analysis, 80.2% of V5-positive cells in contact with an HA-positive cell showed TRACC activation and mCherry expression (n = 106 cells from 50 FOVs).

## Colocalization analysis of sender construct

Sender construct colocalization with endogenous synapsin expression in neurons (*Figure 3—figure supplement 1B-C*) was quantified using the Coloc2 test for image colocalization in Fiji (*Schindelin et al., 2012*), which measures the correlation of pixel intensity at each location to compare HA and synapsin intensity levels.

## Acknowledgements

This work was supported by the NIH (RF1 MH117821 to AYT, DP1NS111132 to MM) and Stanford Wu Tsai Neurosciences Institute (Big Ideas initiative to AYT). KFC was supported by NIH Training Grant 2T32CA009302-41 and the Blavatnik Graduate Fellowship. AYT is an investigator of the Chan Zuckerberg Biohub.

## Additional information

### Funding

| Funder | Grant reference number | Author |
|---|---|---|
| National Institutes of Health | RF1 MH117821 | Alice Y Ting |
| National Institutes of Health | DP1NS111132 | Michelle Monje |
| Stanford Wu Tsai Neurosciences Institute | Big Ideas Initiative | Alice Y Ting |
| National Institutes of Health | 2T32CA009302-41 | Kelvin F Cho |
| Blavatnik Family Foundation | Blavatnik Graduate Fellowship | Kelvin F Cho |
| Chan Zuckerberg Initiative | | Alice Y Ting |

The funders had no role in study design, data collection and interpretation, or the decision to submit the work for publication.

### Author contributions

Kelvin F Cho, Conceptualization, Data curation, Formal analysis, Investigation, Methodology, Validation, Visualization, Writing – original draft, Writing – review and editing; Shawn M Gillespie, Data curation, Formal analysis, Investigation, Methodology, Writing – review and editing; Nicholas A Kalogriopoulos, Investigation, Methodology, Writing – review and editing; Michael A Quezada, Formal analysis, Investigation, Methodology, Writing – review and editing; Martin Jacko, Conceptualization, Writing – review and editing; Michelle Monje, Alice Y Ting, Conceptualization, Funding acquisition, Investigation, Methodology, Project administration, Supervision, Writing – original draft, Writing – review and editing

### Author ORCIDs

Kelvin F Cho ⓘ http://orcid.org/0000-0003-2917-9713
Michelle Monje ⓘ http://orcid.org/0000-0002-3547-237X
Alice Y Ting ⓘ http://orcid.org/0000-0002-8277-5226

### Decision letter and Author response

Decision letter https://doi.org/10.7554/eLife.70881.sa1
Author response https://doi.org/10.7554/eLife.70881.sa2

## Additional files

### Supplementary files

• Transparent reporting form

### Data availability

All data generated or analyzed during this study are included in the manuscript and supporting files.

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
