## [Editor Report]

This manuscript describes engineering of a new system (TRACC) for marking cells that have come into contact with another population of cells. In contrast with previous systems, TRACC is gated temporally and spatially by blue light application. The system comprises a GPCR and ligand that interact at the surface of the two cells, as well as a TEV protease-arrestin fusion that gets recruited following the interaction. The GCPR is fused to a LOV light sensitive domain, a LOV-masked TEV cleavage site and transcriptional activator. TEV cleavage, in the presence of a sender cell and light, releases a transcriptional activator to drive expression of a reporter transgene in the receiver cell. This system provides a new tool for studying cell-cell contacts.

---

## [Decision Letter]

**Decision letter after peer review:**

Thank you for submitting your article "A light-gated transcriptional recorder for detecting cell-cell contacts" for consideration by *eLife*. Your article has been reviewed by 3 peer reviewers, including Volker Dötsch as Reviewing Editor and Reviewer #2, and the evaluation has been overseen by a Reviewing Editor and Philip Cole as the Senior Editor.

Essential revisions:

1) There is no quantification of the detection of neuron-glioma cell interactions in Figure 4 as there is for the other experiments.

2) The authors note that about 80% of cells in contact light up above background, and that some cells not in contact also light up. But "above background" is not defined, nor is what they mean by "some" cells. The data presented in Figure 2E indicate a shift in the distribution as claimed, but a bit more precision in the analysis would be beneficial. It might help to present the data as a violin or box-whisker plot. In addition, the false discovery rate should be assessed.

3) This assay is affinity based, unlike biotin ligase strategies. Therefore, in principle, it might force, rather than detect, interactions between cells. Have the authors seen evidence or such a thing? It would be nice to include data addressing this concern, or cite appropriate evidence assessing this previously with other similar detection strategies.

4) When describing alternative methods that have been published already the authors say that TRACT, transTango, and BAcTrace have been tested only in *Drosophila* cells. This is not necessarily a disadvantage. The authors should say why they think that showing a method for cell-cell contacts in *Drosophila* cells is a disadvantage or remove that statement.

*Reviewer #1:*

Cho and colleagues report the engineering of a light-gated transcriptional reporter of cell-cell contacts (TRACC). The system comprises a GPCR and ligand that interact at the surface of the two cells, as well as a TEV protease-arrestin fusion that gets recruited following the interaction. The GCPR is fused to a LOV light sensitive domain, a LOV-masked TEV cleavage site and transcriptional activator. TEV cleavage, in the presence of a sender cell and light, releases a transcriptional activator to drive expression of a reporter transgene in the receiver cell.

In general, the paper is well-written, and the figures clearly explain the experiments and results.

TRACC is closely related to previous tools from this group, although the permutation described here is novel. It does not seem to perform as well as their other related tools, with higher false positive and false negative rates (~20% each). They also do not really demonstrate the advantage of including a light gate. Although they argue that the light gating provides higher contrast, I can see applications where the requirement for light delivery would make tool use more problematic. There are no real biological applications of the tool presented here.

*Reviewer #2:*

Cho et al., describe the adaption of their previously developed methods SPARK and SPARK2 that detect protein-protein contacts to the detection of cell-cell contacts. The system is elegantly designed and has the advantage that it requires the integration of two independent activation signals. The presence of a ligand for a receptor is not enough, in addition a light stimulus has to occur that results in the proteolytic cleavage of a linker that tethers a transcription factor to the receptor.

The expansion of their previously published method is well designed. The author show the functionality of their system on different cellular systems including HEK293T cells, neurons and neuron-glioma cell contacts.

*Reviewer #3:*

I think this paper describes an interesting method. It also adds an interesting proof-of-principle regarding the detection of neuron-glioma cell interactions. The method is based on a previously described light-gated reporter of GPCR activation (SPARC) combined with a trans-tethered agonist (trans-Tango). In general the data appear solid and interpretation appears reasonable.

Overall I think this is a strong paper, but I do have a couple of specific concerns:

1. The authors note that about 80% of cells in contact light up above background, and that some cells not in contact also light up. But "above background" is not defined, nor is what they mean by "some" cells. The data presented in Figure 2E indicate a shift in the distribution as claimed, but I suggest a bit more precision in the analysis. It might help to present the data as a violin or box-whisker plot. Also, is it possible to include something like a false discovery rate assessment?

2. This assay is affinity based, unlike biotin ligase strategies. Therefore, in principle, it might force, rather than detect, interactions between cells. I don't see any data testing this. Have the authors seen evidence or such a thing? It would be nice to include data addressing this concern, or cite appropriate evidence assessing this previously with other similar detection strategies.

---

## [Author Response]

Essential revisions:1) There is no quantification of the detection of neuron-glioma cell interactions in Figure 4 as there is for the other experiments.

We have repeated this experiment to detect neuron-glioma cell interactions, while including immunostaining for Nestin, a DIPG marker. This allowed us to perform quantification of our results; updated figure panels are now presented in Figure 4C and Figure 4 —figure supplement 1B.

2) The authors note that about 80% of cells in contact light up above background, and that some cells not in contact also light up. But "above background" is not defined, nor is what they mean by "some" cells. The data presented in Figure 2E indicate a shift in the distribution as claimed, but a bit more precision in the analysis would be beneficial. It might help to present the data as a violin or box-whisker plot. In addition, the false discovery rate should be assessed.

We have clarified the text by incorporating details that were previously in the Methods section only. “Above background” means that the mCherry signal in the ROI is > 1.5-fold greater than a blank reference region, and “some cells” refers to the cells that were in contact but did not show mCherry expression (~19.8%, or 21 out of 106 cells). We have also replotted the data in Figure 2E as a violin plot and have included additional clarification regarding false positives/false discovery rate in this assay. Specifically, we explain that about 80% of activated cells are observed to be in contact with a sender cell, while about 20% of activated (mCherry-positive) cells are not in contact with a sender. Based on results of other control experiments (e.g., controls in which sender cells are completely absent or the sender construct is not expressed), the most likely explanation for turn-on in non-contacting cells is that these cells were *previously* in contact with a sender cell, but the sender became dislodged during washing and sample processing. This is explained more clearly in the revised text.

3) This assay is affinity based, unlike biotin ligase strategies. Therefore, in principle, it might force, rather than detect, interactions between cells. Have the authors seen evidence or such a thing? It would be nice to include data addressing this concern, or cite appropriate evidence assessing this previously with other similar detection strategies.

We have not seen any evidence of the receiver and sender constructs driving cell-cell interactions. This contrasts with previous tools we and others have developed, such as split-HRP (Martell et al., *Nature Biotechnology*, 2016) and GRASP (Feinberg et al., *Neuron*, 2008), which *do* drive cell-cell interactions upon overexpression. In the case of TRACC, Figure 2C shows for example that TRACC sender and receiver cells can interact together without exhibiting unnatural morphology. The peptide-GPCR interaction that is the basis of TRACC may be sufficiently low-affinity or transient that it is incapable of significantly perturbing cell-cell contact sites.

4) When describing alternative methods that have been published already the authors say that TRACT, transTango, and BAcTrace have been tested only in *Drosophila* cells. This is not necessarily a disadvantage. The authors should say why they think that showing a method for cell-cell contacts in Drosophila cells is a disadvantage or remove that statement.

We did not intend to convey that a cell-cell contact detection tool in *Drosophila* is disadvantageous, just that the tools have not yet been tested in other species such as mammals or mammalian cell culture. We have removed statements that may be conceived this way and clarified areas in the text in which we mention this.

Reviewer #1:Cho and colleagues report the engineering of a light-gated transcriptional reporter of cell-cell contacts (TRACC). The system comprises a GPCR and ligand that interact at the surface of the two cells, as well as a TEV protease-arrestin fusion that gets recruited following the interaction. The GCPR is fused to a LOV light sensitive domain, a LOV-masked TEV cleavage site and transcriptional activator. TEV cleavage, in the presence of a sender cell and light, releases a transcriptional activator to drive expression of a reporter transgene in the receiver cell.In general, the paper is well-written, and the figures clearly explain the experiments and results.TRACC is closely related to previous tools from this group, although the permutation described here is novel. It does not seem to perform as well as their other related tools, with higher false positive and false negative rates (~20% each). They also do not really demonstrate the advantage of including a light gate. Although they argue that the light gating provides higher contrast, I can see applications where the requirement for light delivery would make tool use more problematic. There are no real biological applications of the tool presented here.

While we report from our HEK293T trans assay specificity and sensitivity values of about 80% each, there are other technical factors in the assay that may affect these numbers. For example, sensitivity may be impacted because the assay is conducted using transient transfection, and for a receiver cell to show TRACC activation, it needs to be simultaneously transfected with 3 separate receiver components (GPCR, arrestin-TEVp, reporter); cells that lack one or more components will decrease sensitivity calculations. In regards to specificity, it is possible that background activation may occur in cells expressing the arrestinTEVp at particularly high levels, which can result in GPCR activation-independent release of the transcription factor. We have also shown previously with SPARK (Kim et al., *eLife*, 2017) that the use of stable cells with more controlled expression of similar components can improve specificity; indeed when we test our DIPG cell lines stably expressing TRACC constructs via transposons, we observe minimal activation in control conditions (Figure 4A, 4C, and Figure 4 —figure supplement 1B). Furthermore, potential dislodging of sender cells during the course of the experiment or during the washing steps in immunostaining can also contribute to the number of (apparent) false positives observed.

Regarding the value and importance of light gating, we show in Figure 1C that omission of the LOV domain results in a drastic increase in background, resulting in little to no ligand-dependent activation. We therefore believe that the light gating is directly responsible for the high SNR of TRACC. It is true that the requirement for light delivery can be limiting for applications in opaque tissue or non-transparent animals where surgical implantation of an optical fiber is not possible.

We show that TRACC can be used successfully to detect cell-cell contacts in both neuron-HEK293T and neuron-glioma co-cultures. We envision that with additional validation of these constructs in vivo, it would be very feasible to use TRACC to detect cellular contacts in mammalian in vivo models.

Reviewer #2:Cho et al., describe the adaption of their previously developed methods SPARK and SPARK2 that detect protein-protein contacts to the detection of cell-cell contacts. The system is elegantly designed and has the advantage that it requires the integration of two independent activation signals. The presence of a ligand for a receptor is not enough, in addition a light stimulus has to occur that results in the proteolytic cleavage of a linker that tethers a transcription factor to the receptor.The expansion of their previously published method is well designed. The author show the functionality of their system on different cellular systems including HEK293T cells, neurons and neuron-glioma cell contacts.

Thank you to the reviewer for the kind comments regarding our work.

Reviewer #3:I think this paper describes an interesting method. It also adds an interesting proof-of-principle regarding the detection of neuron-glioma cell interactions. The method is based on a previously described light-gated reporter of GPCR activation (SPARC) combined with a trans-tethered agonist (trans-Tango). In general the data appear solid and interpretation appears reasonable.Overall I think this is a strong paper, but I do have a couple of specific concerns:1. The authors note that about 80% of cells in contact light up above background, and that some cells not in contact also light up. But "above background" is not defined, nor is what they mean by "some" cells. The data presented in Figure 2E indicate a shift in the distribution as claimed, but I suggest a bit more precision in the analysis. It might help to present the data as a violin or box-whisker plot. Also, is it possible to include something like a false discovery rate assessment?

We have clarified the text by including details that were previously in the Methods section only. “Above background” means that the mCherry signal in the ROI is > 1.5-fold greater than a blank reference region, and “some cells” refers to the cells that were in contact but did not show mCherry expression (~19.8%, or 21 out of 106 cells). We have also replotted the data in Figure 2E as a violin plot and have included additional clarification regarding false positives/false discovery rate in this assay. Specifically, we explain that about 80% of activated cells are observed to be in contact with a sender cell, while about 20% of activated cells are not in contact with a sender. Based on results of other control experiments (e.g., controls in which sender cells are completely absent or the sender construct is not expressed), the most likely explanation for turn-on in non-contacting cells is that these cells were *previously* in contact with a sender cell, but the sender became dislodged during washing and sample processing. This is explained more clearly in the revised text.

2. This assay is affinity based, unlike biotin ligase strategies. Therefore, in principle, it might force, rather than detect, interactions between cells. I don't see any data testing this. Have the authors seen evidence or such a thing? It would be nice to include data addressing this concern, or cite appropriate evidence assessing this previously with other similar detection strategies.

We have not seen any evidence of the receiver and sender constructs driving cell-cell interactions. This contrasts with previous tools we and others have developed, such as split-HRP (Martell et al., *Nature Biotechnology*, 2016) and GRASP (Feinberg et al., *Neuron*, 2008), which *do* drive cell-cell interactions upon overexpression. In the case of TRACC, Figure 2C shows for example that TRACC sender and receiver cells can interact together without exhibiting unnatural morphology. The peptide-GPCR interaction that is the basis of TRACC may be sufficiently low-affinity or transient that it is incapable of significantly perturbing cell-cell contact sites.